# On Analyzing Generative and Denoising Capabilities of Diffusion-based Deep Generative Models

**Kamil Deja**[*][†]
Warsaw University of Technology
Warsaw, Poland
kamil.deja@pw.edu.pl

**Anna Kuzina**[*]
Vrije Universiteit Amsterdam
Amsterdam, the Netherlands
a.kuzina@vu.nl

**Tomasz Trzciński**
Warsaw University of Technology
Jagiellonian University of Cracow
Tooploox, IDEAS NCBR
tomasz.trzcinski@pw.edu.pl

**Jakub M. Tomczak**
Vrije Universiteit Amsterdam
Amsterdam, the Netherlands
j.m.tomczak@vu.nl

## Abstract

Diffusion-based Deep Generative Models (DDGMs) offer state-of-the-art performance in generative modeling. Their main strength comes from their unique setup in which a model (the backward diffusion process) is trained to reverse the forward diffusion process, which gradually adds noise to the input signal. Although DDGMs are well studied, it is still unclear how the small amount of noise is transformed during the backward diffusion process. Here, we focus on analyzing this problem to gain more insight into the behavior of DDGMs and their denoising and generative capabilities. We observe a fluid transition point that changes the functionality of the backward diffusion process from generating a (corrupted) image from noise to denoising the corrupted image to the final sample. Based on this observation, we postulate to divide a DDGM into two parts: a denoiser and a generator. The denoiser could be parameterized by a denoising auto-encoder, while the generator is a diffusion-based model with its own set of parameters. We experimentally validate our proposition, showing its pros and cons.

## 1 Introduction

Diffusion-based Deep Generative Models [22] (DDGM) have recently attracted increasing attention, due to the unprecedented quality of generated samples [5, 9, 11]. The general idea behind this set of methods is to generate samples using diffusion processes [8, 10, 11, 23, 24]. In the *forward diffusion process*, an image is passed through a number of steps that consecutively add a small portion of noise to it. The *backward diffusion process* is a direct reverse of the forward process, where a generative model is trained to gradually denoise the image. With a sufficient number of the forward diffusion steps, noisy images approach isotropic Gaussian noise. Then, generating new examples is possible by applying the backward diffusion to the noise sampled from the standard Gaussian distribution.

While the performance of DDGMs is impressive, not all of their aspects are fully understood. Intuitively, a DDGM is trained to *remove* small amounts of noise from many intermediary corrupted images. Although this perspective is reasonable and complies with the interpretation of DDGMs using stochastic differential equations [10, 24], it is still unclear how the small amount of noise

---

[*]Equal Contribution
[†]Work done while visiting Vrije Universiteit Amsterdam

36th Conference on Neural Information Processing Systems (NeurIPS 2022).

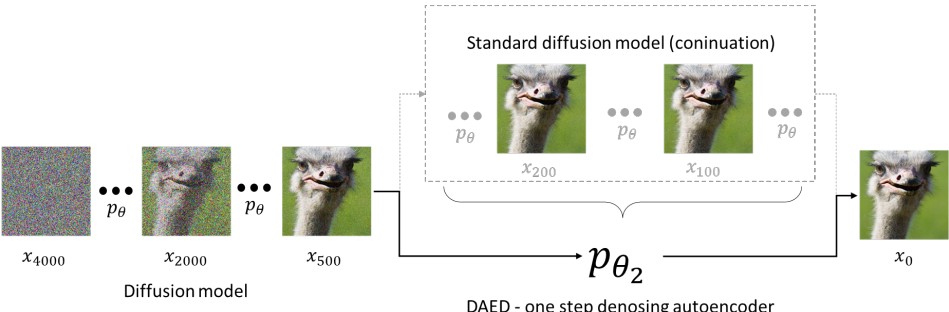

Figure 1: Overview of the proposed Denoising Auto-Encoder with Diffusion (DAED). To validate our hypothesis that DDGMs can be understood as a composition of a *generator* and *denoiser*, we propose to explicitly model the denoising part with a separate denoising autoencoder.

is *removed* during the backward diffusion process where images are composed of almost entirely random values. The more adequate intuition might be that in its initial steps, a diffusion model does not only remove noise but also introduces a new signal according to the distribution learned from the data. In this work, we further investigate this observation to understand the balance between the generative and denoising capabilities of DDGMs.

In particular, we aim to answer the following three questions in this paper: **(i)** Is there a transition in the functionality of the backward diffusion process that switches from generating to denoising? **(ii)** How does this split of functionality affect the performance? **(iii)** Does the denoising part in DDGMs generalize to other data distributions? As a result, the contribution of the paper is threefold:

- First, we analyze the noise distribution in the forward diffusion process and how steps of the diffusion process are correlated with the reconstruction error.
- Second, based on our analysis, we postulate that DDGMs are composed of two parts: a *denoiser* and a *generator*. As a result, we propose a new class of models that consist of a Denoising Auto-Encoder and a Diffusion-based generator shortened as DAED. DAED could be considered as a variation of DDGMs with an explicit split into the denoising part and the generating part.
- Third, we empirically assess the performance of DDGMs and DAED on three datasets (FashionM-NIST, CIFAR10, CelebA) in terms of data generation and transferability (i.e., how DDGMs behave on different data distribution).

## 2   Background

### 2.1   Diffusion-Based Deep Generative Models (DDGMs)

**Model formulation**   We follow the formulation of the Diffusion-based Deep Generative Models (DDGMs) as presented in [8, 22]. DDGMs could be seen as infinitely deep hierarchical VAEs with a specific family of variational posteriors [10, 11, 25, 26], namely, Gaussian diffusion processes [22]. Given a data point $\mathbf{x}_0$ and latent variables $\mathbf{x}_t, \ldots, \mathbf{x}_T$, we want to optimize the marginal likelihood $p_\theta(\mathbf{x}_0) = \int p_\theta(\mathbf{x}_0, \ldots, \mathbf{x}_T) d\mathbf{x}_1, \ldots, \mathbf{x}_T$. We define the *backward* (or *reverse*) *process* as a Markov chain with Gaussian transitions starting with $p(\mathbf{x}_T) = \mathcal{N}(\mathbf{x}_T; \mathbf{0}, \mathbf{I})$, that is:

$$p_\theta(\mathbf{x}_0, \ldots, \mathbf{x}_T) = p(\mathbf{x}_T) \prod_{t=0}^{T} p_\theta(\mathbf{x}_{t-1}|\mathbf{x}_t), \tag{1}$$

where $p_\theta(\mathbf{x}_{t-1}|\mathbf{x}_t) = \mathcal{N}(\mathbf{x}_{t-1}; \mu_\theta(\mathbf{x}_t, t), \Sigma_\theta(\mathbf{x}_t, t))$. Additionally, we define the *forward diffusion process* as a Markov chain that gradually adds Gaussian noise to the data according to a variance schedule $\beta_1, ..., \beta_T$, namely, $q(\mathbf{x}_1, \ldots, \mathbf{x}_T|\mathbf{x}_0) = \prod_{t=1}^{T} q(\mathbf{x}_t|\mathbf{x}_{t-1})$, where $q(\mathbf{x}_t|\mathbf{x}_{t-1}) = \mathcal{N}(\mathbf{x}_t; \sqrt{1 - \beta_t}\mathbf{x}_{t-1}, \beta_t \mathbf{I})$. Let us further define $\alpha_t = 1 - \beta_t$ and $\overline{\alpha}_t = \prod_{i=0}^{t} \alpha_i$. Since the conditionals in the forward diffusion can be seen as Gaussian linear models, we can analytically calculate the following distributions:

$$q(\mathbf{x}_t|\mathbf{x}_0) = \mathcal{N}(\mathbf{x}_t; \sqrt{\overline{\alpha}_t}\mathbf{x}_0, (1 - \overline{\alpha}_t)\mathbf{I}), \tag{2}$$

and

$$q(\mathbf{x}_{t-1}|\mathbf{x}_t, \mathbf{x}_0) = \mathcal{N}(\mathbf{x}_{t-1}; \tilde{\mu}(\mathbf{x}_t, \mathbf{x}_0), \tilde{\beta}_t \mathbf{I}), \tag{3}$$

where $\tilde{\mu}(\mathbf{x}_t, \mathbf{x}_0) = \frac{\sqrt{\bar{\alpha}_{t-1}}\beta_t}{1-\bar{\alpha}_t}\mathbf{x}_0 + \frac{\sqrt{\alpha_t}(1-\bar{\alpha}_{t-1})}{1-\bar{\alpha}_t}\mathbf{x}_t$, and $\tilde{\beta}_t = \frac{1-\bar{\alpha}_{t-1}}{1-\bar{\alpha}_t}\beta_t$. We can use (2) and (3) to define the variational lower bound as follows:

$$\ln p_\theta(\mathbf{x}_0) \geq L_{vlb}(\theta) := \underbrace{\mathbb{E}_{q(\mathbf{x}_1|\mathbf{x}_0)}[\ln p_\theta(\mathbf{x}_0|\mathbf{x}_1)]}_{-L_0} - \underbrace{D_{\mathrm{KL}}\left[q(\mathbf{x}_T|\mathbf{x}_0)\|p(\mathbf{x}_T)\right]}_{L_T}$$

$$-\sum_{t=2}^{T}\underbrace{\mathbb{E}_{q(\mathbf{x}_t|\mathbf{x}_0)}D_{\mathrm{KL}}\left[q(\mathbf{x}_{t-1}|\mathbf{x}_t, \mathbf{x}_0)\|p_\theta(\mathbf{x}_{t-1}|\mathbf{x}_t)\right]}_{L_{t-1}}. \tag{4}$$

that we further optimize with respect to the parameters of the backward diffusion.

**The conditional likelihood** In this paper, we focus on images, thus, data is represented by integers from 0 to 255. Following [8], we scale them linearly to $[-1, 1]$. As a result, to obtain discrete log-likelihoods, we consider the discretized (binned) Gaussian conditional likelihood [8]:

$$p_\theta(\mathbf{x}_0|\mathbf{x}_1) = \prod_{i=1}^{D}\int_{\delta_-(x_0^i)}^{\delta_+(x_0^i)}\mathcal{N}\left(x; \mu_\theta^i(\mathbf{x}_1, 1), \sigma_1^2\right)\mathrm{d}x, \tag{5}$$

where $D$ is the data dimensionality of $\mathbf{x}_0$, and $i$ denotes one coordinate of $\mathbf{x}_0$, and:

$$\delta_+(x) = \begin{cases} \infty & \text{if } x = 1 \\ x + \frac{1}{255} & \text{if } x < 1 \end{cases} \quad \delta_-(x) = \begin{cases} -\infty & \text{if } x = -1 \\ x - \frac{1}{255} & \text{if } x > -1 \end{cases}. \tag{6}$$

**Noise scheduling** Originally, [8] propose to linearly scale the noise parameters $\beta_t$ (*linear scheduling*), e.g., scaling linearly from $\beta_1 = 10^{-4}$ to $\beta_T = 0.02$. In [17], authors suggest to increase the number of less noisy steps through *cosine scheduling*: $\bar{\alpha}_t = \frac{f(t)}{f(0)}, f(t) = \cos\left(\frac{t/T+c}{1+c}\cdot\frac{\pi}{2}\right)^2, c > 0$ with clipping the values of $\beta_t$ to 0.999 to prevent potential instabilities at the end of the diffusion.

**Training details** In [8], authors notice that a single part of the variational lower bound is equal to:

$$L_t(\theta) = \mathbb{E}_{\mathbf{x}_0,\epsilon}\left[\frac{\beta_t^2}{2\sigma_t^2\alpha_t(1-\bar{\alpha}_t)}\left\|\epsilon - \epsilon_\theta\left(\sqrt{\bar{\alpha}_t}\mathbf{x}_0 + \sqrt{1-\bar{\alpha}_t}\epsilon, t\right)\right\|^2\right], \tag{7}$$

where $\epsilon \sim \mathcal{N}(\mathbf{0}, \mathbf{I})$ and $\epsilon_\theta$ is a neural network predicting the noise $\epsilon$ from $\mathbf{x}_t$. Since we use (3) in the variational lower bound objective (4), and $\mathbf{x}_t$ could be sampled from the forward diffusion for a given data, see (2), we can optimize one layer at a time. In other words, we can randomly pick a specific component of the objective, $L_t$, and update the parameters by optimizing $L_t$ without running the whole forward process from $\mathbf{x}_0$ to $\mathbf{x}_T$. As a result, the training becomes very efficient and learning very deep models (with hundreds or even thousands of steps) is possible.

In [8], it is also proposed to train a simplified objective that is a version of (7) without scaling, namely:

$$L_{t,\text{simple}}(\theta) = \mathbb{E}_{\mathbf{x}_0,\epsilon}\left[\left\|\epsilon - \epsilon_\theta\left(\sqrt{\bar{\alpha}_t}\mathbf{x}_0 + \sqrt{1-\bar{\alpha}_t}\epsilon, t\right)\right\|^2\right], \tag{8}$$

where $t$ is uniformly sampled between 1 and $T$. To further reduce computational and memory costs, typically, a single, shared neural network is used for modeling $\epsilon_\theta$ [8, 11, 17] that is parameterized by an architecture based on U-Net type neural net [18]. The U-Net could be seen as a specific auto-encoder that passes all codes from the encoder to the decoder.

## 2.2 Denoising Auto-Encoders

Another class of models, Denoising Auto-Encoders (DAEs), is similar to DDGMs in the sense that they also revert a known corruption process. However, DAEs are trained to remove the noise in a single pass, and unlike DDGMs, they cannot generate new objects. Specifically, DAEs are auto-encoders that reconstruct a data point $\mathbf{x}_0$ from its corrupted (noisy) version [1, 2, 4, 28]. Let us

denote the auto-encoder by $f_\varphi(\cdot)$. Using the same notation as for DDGMs, the Gaussian corruption distribution is $q(\mathbf{x}_1|\mathbf{x}_0)$. Then, a DAE maximizes the following objective function:

$$\ell(\mathbf{x}_0; \varphi) = \mathbb{E}_{q(\mathbf{x}_1|\mathbf{x}_0)} \left[ \ln p\left(\mathbf{x}_0|f_\varphi(\mathbf{x}_1)\right) \right]. \tag{9}$$

and, in particular, for the Gaussian distribution with the identity covariance matrix, we get the original objective for DAEs [28]: $\ln p\left(\mathbf{x}_0|f_\varphi(\mathbf{x}_1)\right) = -\left\|\mathbf{x}_0 - f_\varphi(\mathbf{x}_1)\right\|^2 + const.$

## 3   An analysis of DDGMs

The core idea behind DDGMs is the gradual noise injection to images as we go forward in time such that the final object is a sample from the standard Gaussian distribution. Then, in the backward diffusion process model reverts this procedure and, as a result, generates new objects. Therefore, understanding the success of DDGMs relies heavily on understanding how the injected noise influences the behavior of both training and the model itself.

**The noise distribution in the forward diffusion process**    The first question we ask is how much corrupted an image gets after applying a specific noise schedule. Following [8, 12, 17], we can utilize the signal-to-noise ratio (SNR), expressed as the squared mean of a signal (here: image) divided by the variance of a signal, to quantify the amount of noise in $\mathbf{x}_t$. For this purpose, the quantity of interest is the forward diffusion for a given $\mathbf{x}_0$, namely, $q(\mathbf{x}_t|\mathbf{x}_0)$, that results in the following SNR:

$$SNR(\mathbf{x}_0, t) = \frac{\overline{\alpha}_t \mathbf{x}_0^2}{1 - \overline{\alpha}_t}. \tag{10}$$

Similarly to [11], we formulate the forward diffusion in such a way that the SNR is strictly monotonically decreasing in time, namely, $SNR(\mathbf{x}_0, t) < SNR(\mathbf{x}_0, s)$ for $t > s$. This means that an image becomes more noisy as we go forward in time.

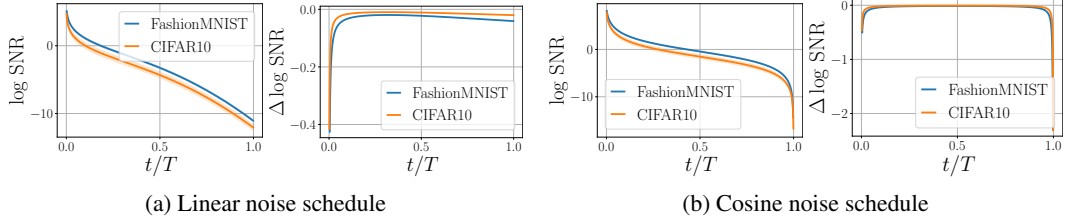

(a) Linear noise schedule                              (b) Cosine noise schedule

Figure 2: Logarithm of the signal-to-noise ratio averaged over the dataset (solid line) and its standard deviation, and the difference of the $\log$ SNR within two consecutive time steps.

In Figure 2 (left) we plot the logarithm of the SNR for both linear (Figure 2.a) and cosine (Figure 2.b) noise schedules for two datasets (FashionMNIST and CIFAR10). We average SNR over the $\mathbf{x}_0$'s (from the corresponding dataset). The right column depicts the change of the $\log$ SNR, i.e., its discrete derivative $\Delta \log \text{SNR}(t) = \log \text{SNR}(x_0, t) - \log \text{SNR}(x_0, t-1)$. First of all, we can notice a point at which the log-SNR drops below $0$. This corresponds to the situation of the noise overshadowing the signal. In the case of the linear noise schedule, this happens after about $20\%$ of steps, while for the cosine noise schedule, it appears after about $25 - 50\%$ of steps. However, the transition occurs in both cases. The biggest changes in the log-SNR are noticeable within the first $10\%$ of steps. This may suggest that the signal is the strongest within the first $10 - 20\%$ of the forward diffusion process steps, and then it starts being overshadowed by the noise.

**The reconstruction error of DDGMs**    Since we know that the signal is not lost within the first $10 - 20\%$ of steps, the next question is about the reconstruction capabilities of DDGMs, namely, what is the reconstruction error of $\mathbf{x}_t \sim q(\mathbf{x}_t|\mathbf{x}_0)$. To be clear, we are not interested in how much each step of a DDGM contributes to the final objective (e.g., see Figure 2 in [17]) but rather how well a DDGM reconstructs a noisy image $\mathbf{x}_t$. In Figure 3 we plot the *Mean Absolute Error* (MAE) and the *Multi-Scale Structural Similarity* (MS-SSIM) [29] that both measure the difference between an original image $\mathbf{x}_0$ and a corrupted image at the $t^{th}$ step $\mathbf{x}_t$ reversed by the backward diffusion.

We present the values on two datasets (FashionMNIST and CIFAR10) for the first 20% of steps. Apparently, after around 10% of the steps, the reconstruction error starts growing, and the MAE increases linearly above 0.1 (i.e., about 6% of error per pixel). At the same time, the MS-SSIM drops below $0.9 - 0.95$ (i.e., the discrepancy between original images and reconstructions becomes perceptually evident). This observation might suggest that DDGMs could be roughly

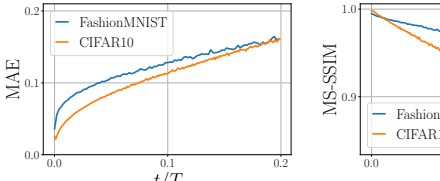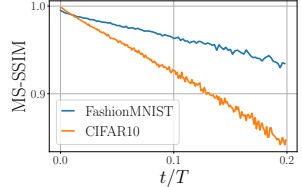

Figure 3: The averaged reconstruction error calculated using (*left*) the MAE, and (*right*) the MS-SSIM at different steps of a DDGM.

divided into two parts: a fraction of steps of a DDGM (e.g., first 10% of the steps) constitute a *denoiser* that turns a corrupted image into a clear image, and the remaining steps of the DDGM are responsible for turning noise into a noisy structure (a corrupted image), i.e., a *generator* that generates meaningful patterns. In other words, we claim that DDGM can be interpreted as a composition of a denoiser and a generator, but the boundary between those two parts is fluid. Moreover, the denoiser gradually removes the noise in a generative manner (i.e., by sampling $\mathbf{x}_{t-1} \sim p(\mathbf{x}_{t-1}|\mathbf{x}_t)$).

**DDGMs as hierarchical VAEs** In this paper, we postulate that DDGMs could be seen as a composition of parts that serve different purposes. We can get additional insight into our claim by noticing a close connection between DDGMs and hierarchical VAEs. As presented in [10, 11, 25], if we treat all $\mathbf{x}_t$'s with $t > 0$ as latents, and see the forward diffusion process as a composition of (non-trainable) variational posteriors, DGGMs become a specific formulation of hierarchical VAEs. On the other hand, we can start with a VAE with a single latent variable, $\mathbf{x}_1$, for which the variational lower bound is equal to:

$$\ln p(\mathbf{x}_0) \geq \mathbb{E}_{\mathbf{x}_1 \sim q(\mathbf{x}_1|\mathbf{x}_0)} \left[ \ln p(\mathbf{x}_0|\mathbf{x}_1) \right] - D_{\text{KL}}[q(\mathbf{x}_1|\mathbf{x}_0)||p(\mathbf{x}_1)]. \tag{11}$$

Then, similarly to [27, 30], the marginal $p(\mathbf{x}_1)$ could be further modeled by a DDGM. By keeping the dimensionality of $\mathbf{x}_1$ the same as $\mathbf{x}_0$, and taking the variational posterior $q(\mathbf{x}_1|\mathbf{x}_0)$ to be fixed and part of the forward diffusion, we get the DDGM model. This perspective of combining a VAE with a DDGM opens new possibilities for developing hybrid models.

## 4 DAED: Denoising Auto-Encoder with Diffusion

In this work, we propose a specific combination that distinctly splits the DDGM into generative and denoising parts. As noted in the previous section, the signal in the forward diffusion process is the strongest within the first $10 - 20\%$ of steps, and, thus, we postulate to perceive this first part of a DDGM as a denoiser. Together with the observation about the combination of a VAE with a DDGM-based prior, we consider turning a denoising auto-encoder into a generative model as presented in Figure 1. We bring a DDGM-based part into DAE for generating corrupted images. The resulting objective is the following:

$$\bar{\ell}(\mathbf{x}_0; \varphi, \theta) = \mathbb{E}_{\mathbf{x}_1 \sim q(\mathbf{x}_1|\mathbf{x}_0)} \left[ \ln p\left(\mathbf{x}_0|f_\varphi\left(\mathbf{x}_1\right)\right) + \ln p_\theta(\mathbf{x}_1) \right] \tag{12}$$

$$\geq \underbrace{\mathbb{E}_{\mathbf{x}_1 \sim q(\mathbf{x}_1|\mathbf{x}_0)} \left[ \ln p\left(\mathbf{x}_0|f_\varphi\left(\mathbf{x}_1\right)\right) \right]}_{\ell_{\text{DAE}}(\mathbf{x}_0; \varphi)} + \underbrace{\mathbb{E}_{q(\mathbf{x}_2,\ldots,\mathbf{x}_T|\mathbf{x}_1)} \left[ \frac{\ln p_\theta(\mathbf{x}_1, \ldots, \mathbf{x}_T)}{q(\mathbf{x}_1, \ldots, \mathbf{x}_T|\mathbf{x}_0)} \right]}_{\ell_{\text{D}}(\mathbf{x}_0; \theta)}, \tag{13}$$

where in (13) we introduce additional latent variables and the variational posterior over them, that yields the variational lower bound. We call the resulting model *DAE with a Diffusion*, or DEAD for short. In a sense, DAED is a DDGM with distinct parameterizations of the part between $\mathbf{x}_0$ and $\mathbf{x}_1$, and the part for the remaining $\mathbf{x}$'s. Thus, DEAD is almost identical to a DDGM, but there are the following differences: **(i)** We can control the amount of noise in $q(\mathbf{x}_1|\mathbf{x}_0)$. It can correspond to the first step of the forward diffusion model, or we can introduce more noise at once that would correspond to several steps in the DDGM. **(ii)** We use two different parameterizations, namely, an auto-encoder (e.g., a U-Net architecture) for $f_\varphi(\cdot)$ and a separate, shared U-Net for modeling the DDGM from $\mathbf{x}_1$ to $\mathbf{x}_T$. Since there are two neural networks, the lower bound to the objective $\bar{\ell}$ is in fact a composition of two objectives with disjunctive parameters, namely, the objective for the

*denoiser*, $\ell_{\text{DAE}}$, and the objective for the *generator* (i.e., the diffusion-based generative model), $\ell_{\text{D}}$. **(iii)** In the DAED, we introduce the *denoiser* explicitly and make a clear distinction between the denoising and the generating parts while, as discussed earlier, this boundary is rather fluid in DDGMs. By introducing DAED, we can analyze what happens if we distinctly divide those two aspects with two separate parametrizations.

Moreover, we hypothesize that the resulting model may better generalize across various data distributions due to decoupling the parameterization of the denoiser and the generator. The training dataset may bias a single, shared parameterization in a DDGM, and while denoising an image from a different domain, it may add some artifacts from the source. While with two distinct parameterizations, there might be a lower chance for that. We evaluate this hypothesis in the experiments.

## 5 Related work

**DDGM for image generation** Various modifications of DDGMs were recently proposed to improve their sampling quality. This includes simplifying the learning objective and proposing new noise schedulers, which allow DDGMs to achieve state-of-the-art results. In this work, we show that splitting the decoder into two parts, namely, a denoiser and a generator, can benefit the performance, especially when training with the variational lower bound.

**Properties of DDGMs** In [8] authors notice that DDGMs can be beneficial for lossy compression, observing (Figure 5 in [8]) that most of the bits are allocated to the region of the smallest distortion that corresponds to the first steps of a DDGM. We draw a similar conclusion when discussing the denoising ability of the diffusion model in Section 3. However, we base our analysis on the signal-to-noise ratio rather than compression. On the other hand [21] focus on the computational complexity of DDGM and propose a progressive distillation that iteratively reduces the number of diffusion steps. The work shows that it is possible to considerably reduce the number of sampling steps without losing performance. We believe that their results support our intuition that it is reasonable to combine several initial steps into a single denoiser model. In [3], authors evaluate how the diffusion process changes in time when model is trained with different objectives (Eq. 7 or Eq. 8). They observe that the image generation process differs significantly and that it is more beneficial to switch between those two approaches at different stages of the diffusion. In this work, we also investigate changes in the diffusion process, but we focus on the generative and denoising capabilities of the model instead.

**Connection to hierarchical Variational Autoencoders** Several works have noted the connection of DDGM to VAEs. In [10] authors focus on the continuous diffusion models and draw the connection to the infinitely deep hierarchical VAEs. In [11] authors further explore this connection, formulate a VLB objective in terms of the signal-to-noise ratio and propose to learn noise schedule, which brings the forward diffusion process even closer to the encoder of a VAE. Recently a latent score-based generative model (LSGM) was proposed [27], which can be seen as a VAE with the score-based prior. We follow a similar direction and propose to see a DDGM as a combination of a denoising auto-encoder with an additional diffusion-based generator of corrupted images.

## 6 Experiments

**Experimental setup** In all the experiments, we use a U-Net-based architecture with timestep embeddings as proposed in [8, 17]. We train all the models with a linear $\beta$ scheduler and uniform steps sampler to simplify the comparison. All implementation details and hyperparameters are included in the Appendix ( A.4) and code repository [3]. For DAED, we use the same architecture for both the diffusion part and the denoising autoencoder. We run experiments on three standard benchmarks with different complexity: FashionMNIST [31] of gray-scale $28 \times 28$ images, CIFAR-10 [13] of $32 \times 32$ natural images, and CelebA [14] of $64 \times 64$ photographs of faces. We do not use any augmentations during training for any dataset. We report results for both variational lower bound loss (VLB) [22] and simplified objective [8]. Following [17] we evaluate the quality of generations with Fréchet Inception Distance (FID) [6] and distributions Precision (Prec) and Recall (Rec) metrics [19] that disentangle FID score into two aspects: the quality of generated results (Precision) and their diversity (Recall).

---

[3]`https://github.com/KamilDeja/analysing_ddgm`

## 6.1 Is there a transition in functionality of the backward diffusion process that switches from generating to denoising?

In section 3, we investigate how the signal-to-noise ratio and the reconstruction error of a DDGM change with the increasing number of diffusion steps (see Figure 4). Based on this analysis, we postulate that DDGMs can be divided into two parts: a *denoiser* and a *generator*. To determine the switching point, we propose an experiment that answers the following question:

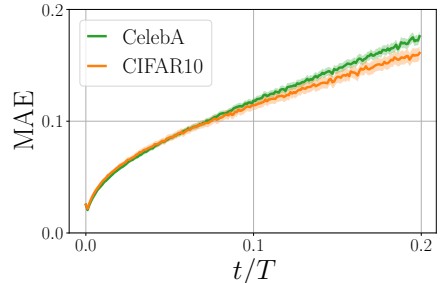

Figure 4: The MAE for a DDGM trained on CIFAR10 and evaluated on CIFAR10 & CelebA, with a 0.95 confidence interval.

*Is there a denoising part of a DDGM that is agnostic to the signal from the data?*

To that end, we refer once more to the analysis of the reconstruction error (e.g., MAE) from different diffusion steps. This time, however, we compare the quality of reconstructions with a single DDGM model trained on the CIFAR10 dataset and then evaluated on CIFAR10 and CelebA. The result of this experiment is presented in Figure 4. Interestingly, we notice that for approximately $10\%$ of the initial steps of the DDGM, there is a negligible difference in the reconstruction error between these two datasets. This fact may suggest that, indeed, the model does not require any information about the background data signal in the first steps, and it is capable of denoising corrupted images. However, after this point (about $10\%$ of steps), the reconstruction error starts growing faster for the dataset the model was not trained on. This indicates that information about the domain becomes important and affects performance.

## 6.2 How does splitting DDGMs into generative and denoising parts affect the performance?

The results so far confirm our claims that DDGMs could be divided into denoising and generative parts. Independently of a dataset, there appears to be a transition point at which a DDGM stops generating a corrupted image from noise and starts denoising it in a generative manner. Here, we aim to verify whether it is possible to do a clear split into a denoising part and a generating part. For this purpose, we use the introduced DAED approach that consists of a DAE part (the denoiser) and a DDGM (the generator) parameterized by two distinct U-Nets.

First, we consider a situation in which we train a DDGM using the simplified objective (8) and then replace the first steps with a DAE. In other words, we train a DAED in two steps: first the DDGM and then the DAE. This experiment aims to check how the decoupling of the DDGM into two parts influences the model performance. In Figure 5 we present the dependency between the log-SNR at the splitting point and the FID score. In all cases, the performance of DAED is comparable to the DDGM if we replace the DAE with up to the $10\%$ of the steps that correspond to $log(SNR)$ is equal to around $4$. For more complicated datasets like CIFAR10 and CelebA, fewer steps could be replaced. This effect could be explained by the fact that images in these datasets have three channels (RGB), and removing noise is more problematic. That outcome reconfirms our presumptions that it is reasonable to split the DDGM since the final performance is not significantly affected by the division for an adequately chosen splitting point.

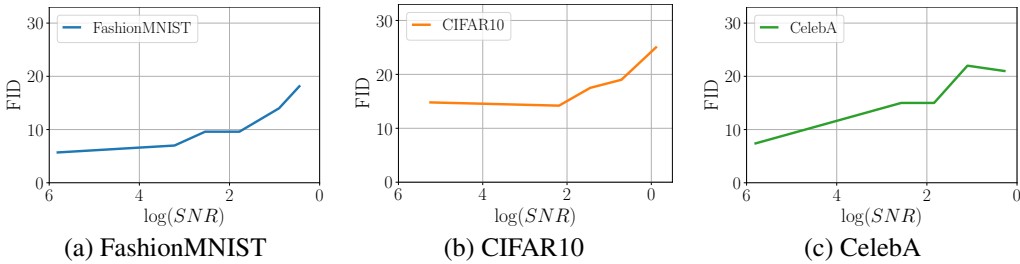

|  (a) FashionMNIST | (b) CIFAR10 | (c) CelebA |

Figure 5: The performance (FID) of DAED with different switching points with respect to the logarithm of the signal to noise ratio (10) on three different datasets.

Table 1: FID Precision (Prec) and Recall (Rec) scores. For each row, we indicate the length of the diffusion process (T) and the training objective (Loss). **Best results in bold**.

| Model | | | Fashion Mnist | | | | CIFAR10 | | | | CelebA | | |
|---|---|---|---|---|---|---|---|---|---|---|---|---|---|
| | Loss | | T | FID ↓ | Prec ↑ | Rec ↑ | T | FID ↓ | Prec ↑ | Rec ↑ | T | FID ↓ | Prec ↑ | Rec ↑ |
| DDGM | VLB | | 500 | 8.9 | 68 | 53 | 1000 | 26 | 53 | 54 | 1000 | 23 | 51 | 21 |
| DAED $\beta_1 = 0.1$ | VLB | | 468 | 9.1 | **71** | 60 | 900 | 20 | 59 | 46 | 900 | 18 | 63 | **30** |
| DAED $\beta_1 = 0.001$ | VLB | | 499 | **7.5** | **71** | 64 | 999 | **15** | **60** | **60** | 999 | **16** | **70** | 27 |
| DDGM | Simple | | 500 | 7.8 | 72 | **65** | 1000 | **7.2** | 65 | 61 | 1000 | **4.9** | 66 | **57** |
| DAED $\beta_1 = 0.1$ | Simple | | 468 | 9.6 | **73** | 58 | 900 | 19 | 62 | 50 | 900 | 22 | **67** | 27 |
| DAED $\beta_1 = 0.001$ | Simple | | 499 | **5.7** | 69 | 64 | 999 | 14.8 | 65 | 53 | 999 | 7.4 | **67** | 54 |

To get further insight into the qualitative performance, in Figure 6 we demonstrate how the selection of the splitting point with respect to the Signal to Noise Ratio (SNR) affects the quality of final generations[4]. We present non-cherry-picked samples from DAED trained in the same manner as described in the previous paragraph. As expected, the more noise the DAE part (the denoiser) must deal with (see the values of $\beta_1$ in Figure 6), the fewer details in the generations there are. These samples again indicate that by replacing some steps with a denoiser, we get a trade-off between "cleaning" the corrupted image or, in fact, further generating details. It seems that there is a sweet spot for

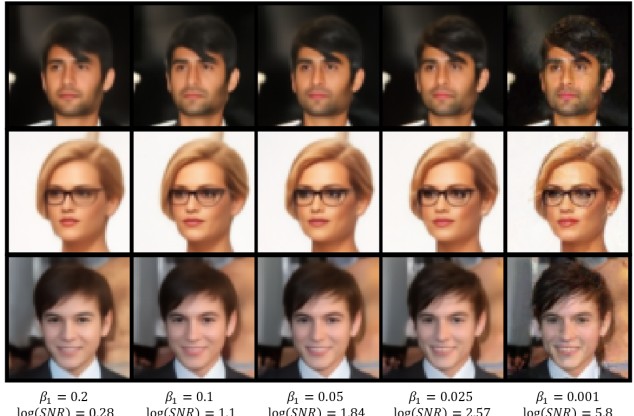

$\beta_1 = 0.2$  $\beta_1 = 0.1$  $\beta_1 = 0.05$  $\beta_1 = 0.025$  $\beta_1 = 0.001$
$\log(SNR) = 0.28$  $\log(SNR) = 1.1$  $\log(SNR) = 1.84$  $\log(SNR) = 2.57$  $\log(SNR) = 5.8$

Figure 6: Examples of generations from DAED with the same noise value and different switching points.

perceptually appealing images that contain details and are "smooth" at the same time, see $\beta_1 = 0.025$ in Figure 6. However, as it is typically difficult to provide convincing arguments by *staring* at samples, we further propose to analyze quantitative measures.

In Table 1, we compare the performance of DAED against the DDGM on FashionMNIST, CIFAR10, and CelebA in terms of FID, Precision and Recall scores. We want to highlight that our goal is not to achieve SOTA results on the before-mentioned datasets but to verify whether we can gain some further understanding and, potentially, some improvement by splitting the denoising and generative parts. We consider two scenarios, namely, learning a DDGM and DAEDs using either the variational lower bound (VBL) or the simplified objective (Simple) with various lengths of the diffusion. Interestingly, DAED outperforms the DDGM when these models are trained using the VBL loss. For the simplified objective, DAED trained with the same number of diffusion steps yields slightly lower performance than standard DDGMs. As indicated by the Precision/Recall, generations from DAED are as precise as those from DDGM. However, they lack certain diversity, probably due to the smoothing effect of the DAE part. Detailed results for other setups are presented in Appendix A.[5]

## 6.3 Does the noise removal in DDGMs generalize to other data distributions?

The last question we are interested in is the generalizability of DDGMs to other data distributions. We refer to this concept as *transferability* for short. In other words, the goal of this experiment is to determine whether we can reuse a model or its part on new data with as good performance as possible. In this experiment, we rely on the results presented in Section 6.1 where roughly the first 10% of steps could be seen as the denoising part. To further strengthen this perspective, we also utilize DAED with an explicit division into the denoising and generating parts.

First, we consider the case in which we compare the reconstruction errors measured by the MAE and the MS-SSIM. In this scenario, we train a DDGM on a source dataset and then assess it on

---

[4]Generations for all datasets are presented in Appendix A.2
[5]In Appendix A.6 we show that increasing the number of parameters of DDGMs to be comparable to DAED does not lead to significant performance improvements.

Table 2: Reconstruction errors measured by MAE ($\downarrow$), MS-SSIM ($\uparrow$) for images noised with $\beta_1 = 0.1$. *To evaluate models trained on CIFAR10, we downscale CelebA to $32 \times 32$. **Best results in bold.**

| | Target dataset | CIFAR10 | | CIFAR100 | | CelebA* | |
|---|---|---|---|---|---|---|---|
| Source Dataset | Model | MAE | MS-SSIM | MAE | MS-SSIM | MAE | MS-SSIM |
| CIFAR10 | DDGM VLB | 0.091 | 0.94 | 0.097 | 0.94 | 0.093 | 0.95 |
| | DDGM Simple | 0.085 | 0.95 | 0.097 | 0.94 | 0.096 | 0.95 |
| | DAED | **0.065** | **0.97** | **0.074** | **0.97** | **0.068** | **0.97** |
| ImageNet | DDGM VLB | 0.113 | 0.93 | 0.110 | 0.93 | 0.077 | 0.96 |
| | DDGM Simple | 0.113 | 0.94 | 0.111 | 0.93 | 0.068 | 0.96 |
| | DAED | **0.071** | **0.97** | **0.071** | **0.97** | **0.050** | **0.98** |

a target dataset. We use CIFAR10 or ImageNet (32x32 or 64x64) as source data and CIFAR10, CIFAR100, or CelebA as target data. For each image from the target dataset, we apply the DAE part of DAED to obtain the reconstruction or 793 steps of the forward and backward diffusion in the case of the DDGM, which corresponds to the same level of added noise. For this experiment, we use the pre-trained DDGM from [17] that consists of 4000 steps and uses the cosine noise scheduler. The results are outlined in Table 2. First of all, there is no significant difference in the performance of DDGMs trained with either the VBL objective or the simplified objective. They achieve a quite satisfactory MAE and MS-SSIM scores. However, DAED outperforms the DDGMs, obtaining much better transferability. We explain it by the fact that probably, with each step in the denoising part DDGM adds details that are typical for source data while DAED focuses on removing noise and produces a smoother output. This outcome may further suggest that splitting DDGMs into two parts with two separate parameterizations is reasonable and even beneficial.

To get further insight into the transferability behavior, we present a few (non-cherry-picked) examples from CelebA in Figure 7.a and four toy examples in Figure 7.b. We use the same setup as explained in the previous paragraph (i.e., the pre-trained DDGM provided in [17]), and the images are noised with $\beta_1 = 0.1$. In columns 3–6 in Figure 7.a, we present reconstructions for the DDGM trained on CelebA, the DDGM trained on ImageNet, DAED trained on CelebA, and DAED trained on ImageNet, respectively. It becomes apparent that the DDGM trained on CelebA denoises the image by generating new details while DAED denoises by smoothing. Interestingly, DAED performs better than the DDGM when we use ImageNet-trained models to denoise CelebA. In Figure 7.b, we depict several toy examples that were denoised with the DDGM and DAED trained on CIFAR10. We see that the DDGM adds many details that are artifacts from the source data. It seems that DAED does not suffer from that behavior.

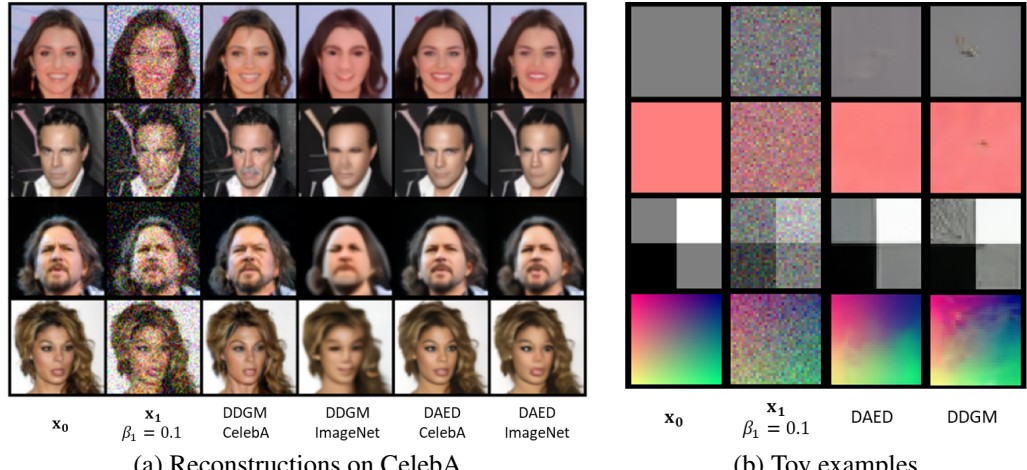

(a) Reconstructions on CelebA      (b) Toy examples

Figure 7: (a) Denoising of image with $0.1$ noise using either DAED or the corresponding number of the DDGM steps. (b) Four noisy toy examples denoised by DAED and the DDGM.

## 7 Conclusion

In this work, we investigate the generative and denoising capabilities of the Diffusion-based Deep Generative Models. We observe and experimentally validate that it is reasonable to understand DDGMs as a combination of two parts. The first one generates noisy samples from the pure noise by inputting more signal from a learned data distribution, while the second one removes the remaining noise from the signal. Although for standard DDGMs, the exact switching point between those two parts is fluid, we propose a new approach dubbed DAED that is explicitly built as a combination of a generative component (a DDGM) and a denoising one (a DAE). In the experiments, we observe that DAED simplifies training with a standard VLB loss function that leads to improved performance. On the other hand, with increasing noise processed by DAE, DAED smoothens the generations resulting in lower performance when training with the simplified objective. We further show that DDGMs, and DAED especially, generalize well to unseen data, what opens new possibilities for further research in terms of transfer or continual learning of DDGMs.

## 8 Acknowledgements

This research was funded by National Science Centre, Poland (grant no 2018/31/N/ST6/02374 and 2020/39/B/ST6/01511), Foundation for Polish Science (grant no POIR.04.04.00-00-14DE/18-00 carried out within the Team-Net program co-financed by the European Union under the European Regional Development Fund), and the Hybrid Intelligence Center, a 10-year programme funded by the Dutch Ministry of Education, Culture and Science through the Netherlands Organisation for Scientific Research. Computation was carried out on the Dutch national e-infrastructure with the support of SURF Cooperative.

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
