# A  Additional experiments

In this section, we present extended evaluation of all models introduced in the main work. Following [17], we show the assessment of generations quality in terms of additional metrics namely Inception Score [20] and spatial Fréchet Inception Distance [16] – a version of standard FID score but based on spatial image features.

Table 3: Extended evaluation results for CIFAR10 dataset.

| Model | | | CIFAR-10 | | | | |
|---|---|---|---|---|---|---|---|
| | Loss | T | IS ↑ | FID ↓ | sFID ↓ | Prec ↑ | Rec ↑ |
| DDGM | VLB | 1000 | 7.6 | 26.1 | 10.5 | 54 | 55 |
| DAED $\beta_1 = 0.1$ | VLB | 900 | 8.2 | 20.4 | 16.1 | 59 | 46 |
| DAED $\beta_1 = 0.025$ | VLB | 979 | 7.7 | 22.4 | 15.8 | 57 | 53 |
| DAED linear | VLB | 999 | 8.1 | 14.5 | 9.8 | 60 | 59 |
| DDGM | Simple | 1000 | 9.5 | 7.2 | 8.6 | 65 | 61 |
| DAED $\beta_1 = 0.2$ | Simple | 891 | 7.8 | 29.4 | 24.7 | 53 | 40 |
| DAED $\beta_1 = 0.1$ | Simple | 900 | 8.0 | 19.0 | 14.9 | 62 | 50 |
| DAED $\beta_1 = 0.025$ | Simple | 979 | 8.6 | 14.2 | 14.6 | 60 | 53 |
| DAED $\beta_1 = 0.001$ | Simple | 999 | 9.1 | 14.9 | 10.1 | 66 | 54 |

Table 4: Extended evaluation results for CelebA dataset. Additionally to standard models, we also include evaluation for DAED setup where DAE model is trained only on ImageNet dataset.

| Model | | | CelebA | | | | |
|---|---|---|---|---|---|---|---|
| | Loss | T | IS ↑ | FID ↓ | sFID ↓ | Prec ↑ | Rec ↑ |
| DDGM | VLB | 1000 | 2.4 | 23.1 | 37.3 | 51 | 21 |
| DAED $\beta_1 = 0.1$ | VLB | 900 | 2.9 | 18.2 | 23.9 | 63 | 31 |
| DAED $\beta_1 = 0.025$ | VLB | 979 | 2.7 | 25.4 | 35.8 | 64 | 17 |
| DAED linear | VLB | 1000 | 2.6 | 16.8 | 23.6 | 70 | 27 |
| DDGM | Simple | 1000 | 3.0 | 6.1 | 14.7 | 66 | 56 |
| DAED $\beta_1 = 0.2$ | Simple | 890 | 2.7 | 21.0 | 31.2 | 63 | 22 |
| DAED $\beta_1 = 0.1$ | Simple | 900 | 3.0 | 17.0 | 23.3 | 66 | 31 |
| DAED $\beta_1 = 0.025$ | Simple | 979 | 2.7 | 15.1 | 17.6 | 64 | 38 |
| DAED $\beta_1 = 0.001$ | Simple | 999 | 2.8 | 6.2 | 11.0 | 69 | 55 |
| DAED (IN) $\beta_1 = 0.1$ | Simple | 900 | 2.9 | 25.6 | 30.5 | 44 | 29 |

Table 5: Extended evaluation results for Fashion MNIST dataset.

| | | | Fashion Mnist | | | | |
|---|---|---|---|---|---|---|---|
| | Loss | T | IS ↑ | FID ↓ | sFID ↓ | Prec ↑ | Rec ↑ |
| DDGM | vlb | 500 | 4.1 | 8.9 | 11 | 68 | 53 |
| DAED $\beta_1 = 0.1$ | vlb | 468 | 4.06 | 9.1 | 13 | 71 | 60 |
| DAED $\beta_1 = 0.025$ | vlb | 489 | 4.02 | 9.7 | 11 | 70 | 62 |
| DAED linear | vlb | 499 | 4.1 | 7.5 | 11.3 | 70.5 | 64 |
| DDGM | Simple | 500 | 4.3 | 7.8 | 9.03 | 71.5 | 65.3 |
| DAED $\beta_1 = 0.3$ | Simple | 426 | 3.78 | 18 | 24 | 73.8 | 41 |
| DAED $\beta_1 = 0.2$ | Simple | 445 | 3.87 | 14 | 20 | 74.8 | 47 |
| DAED $\beta_1 = 0.1$ | Simple | 468 | 3.95 | 9.6 | 11.2 | 73.2 | 58.4 |
| DAED $\beta_1 = 0.025$ | Simple | 489 | 4.05 | 7.36 | 13 | 73 | 61 |
| DAED $\beta_1 = 0.001$ | Simple | 499 | 4.3 | 5.7 | 11.3 | 69.3 | 64.2 |

## A.1 Signal-to-noise ratio detailed plots

In this section we present detailed signal-to-noise ratio (SNR) plots that are used for analysis in Sec. 3 for all evaluated datasets. Independently on the original dataset, SNR changes in the similar manner – with the most drastic loss in the first 10% steps.

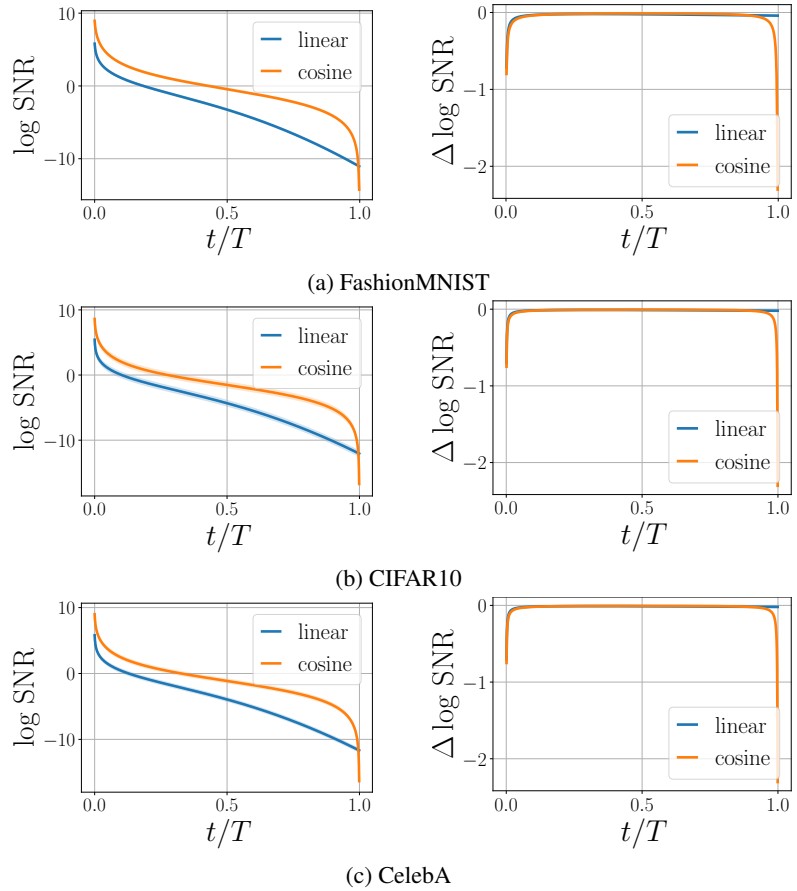

(a) FashionMNIST

(b) CIFAR10

(c) CelebA

Figure 8: Signal-to-noise ratio and its discrete derivative for each of the three datasets: (a) Fashion-MNIST, (b) CIFAR10 and (c) CelebA).

## A.2 Examples of generations

In this section we present generations for all datasets with different models we compare in this work.

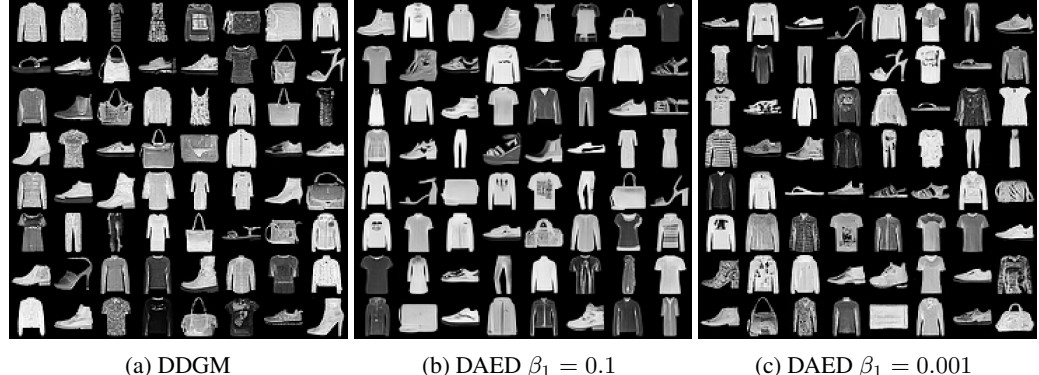

(a) DDGM       (b) DAED $\beta_1 = 0.1$       (c) DAED $\beta_1 = 0.001$

Figure 9: Generations from different models trained on FashionMNIST dataset. All models were trained with Simple loss function.

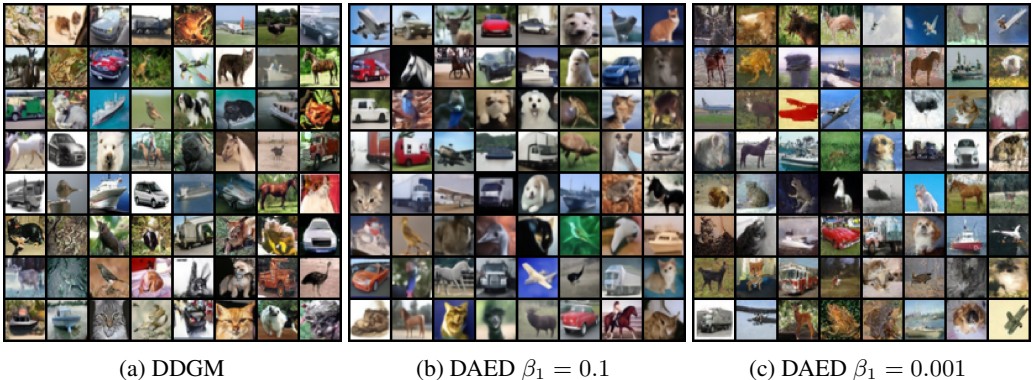

(a) DDGM       (b) DAED $\beta_1 = 0.1$       (c) DAED $\beta_1 = 0.001$

Figure 10: Generations from different models trained on CIFAR10 dataset. All models were trained with Simple loss function.

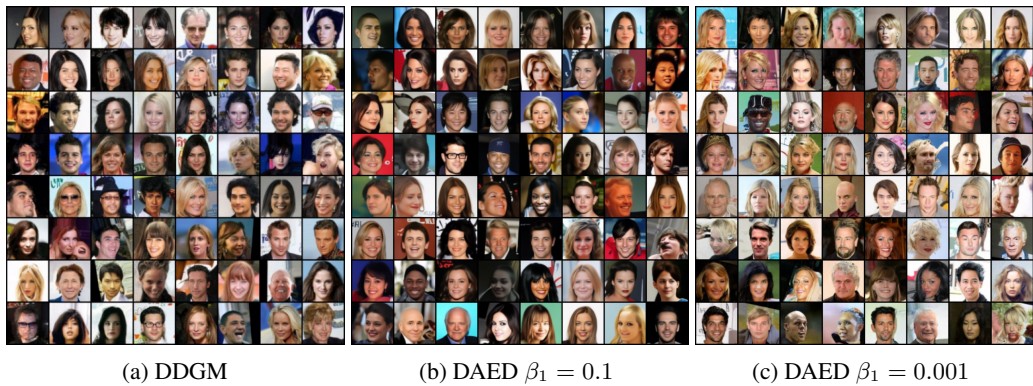

(a) DDGM       (b) DAED $\beta_1 = 0.1$       (c) DAED $\beta_1 = 0.001$

Figure 11: Generations from different models trained on CelebA dataset. All models were trained with Simple loss function.

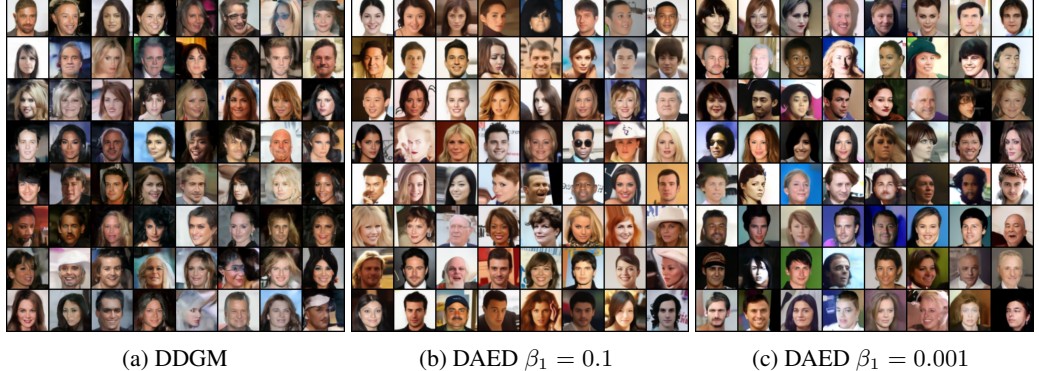

| (a) DDGM | (b) DAED $\beta_1 = 0.1$ | (c) DAED $\beta_1 = 0.001$ |

Figure 12: Generations from different models trained on CelebA dataset with original VLB loss function.

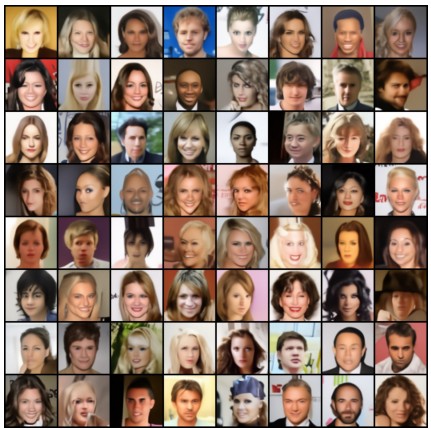

(a) DAED $\beta_1 = 0.1$

Figure 13: Generations from DAED model where DDGM part was trained on CelebA dataset while DAE on ImageNet.

### A.3 Training Dynamics

**How does the objective of a diffusion model change in time?**  In the standard DDGM setup, a single model is optimized with a joint loss from all of the diffusion steps. However, as depicted in Fig 15a, different parts of the diffusion contribute to the sum differently. In fact, the first step of the diffusion is already responsible for 75% of the whole training loss, while first 1% of steps contributes to the over the 90% of the training objective. This observation implies that a single neural network applied to all diffusion steps is mostly optimized to denoise the initial steps. In Fig. 14 we present how this loss contribution changes over time. Surprisingly, only 2% of the training time is needed to align latter 90% of training steps to the loss value below $0.01$. These observations led to the emergence of cosine scheduler [17] where authors change the noise scheduler to increase the number of steps with higher loss values.

In this work, we propose to tackle this problem from a different perspective and to analyze what happens if we detach the loss from initial diffusion steps from the total sum. In Figure 15b, we compare how such a detachment of the first step of 1000-stepped DDGM with DAED influence the loss value on the remaining 999 steps. As depicted in DAED, the loss converges to lower values that explains the improvement of the performance of DAED when training with the VLB loss.

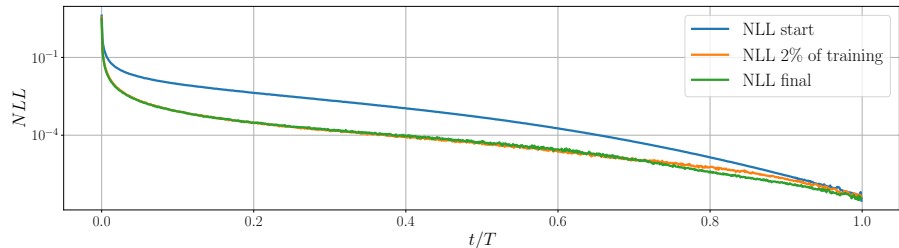

Figure 14: Dynamics of the negative log likelihood for different steps of standard DDGM trained on CIFAR10 with VLB objective. Already after 2% of training time, $p_\theta$ converges to very low loss values (below 0.001) for all of the training steps above 0.1T.

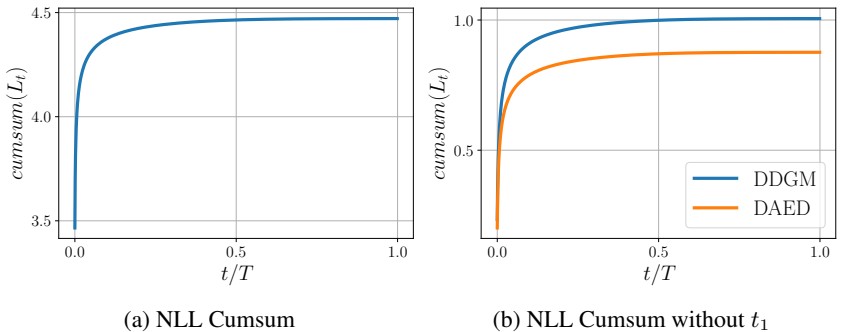

(a) NLL Cumsum

(b) NLL Cumsum without $t_1$

Figure 15: The cumulative sum of the negative log likelihood for different steps of a standard DDGM trained on CIFAR10 with the VLB objective (left), and the same cumulative sum without the first diffusion step in comparison to DAED with exactly the same $\beta$ scheduler.

## A.4 Training Hyperparameters

In all of our experiments, we follow [17]. We train all models with U-Net architecture, with three or four depth levels (depending on a dataset), with three residual blocks each, with a given number of filters depending on the dataset – as presented in 6. In all of our models, we use time embeddings and attention-based layers with three attention heads in each model.

We optimize our models on the basis of randomly selected diffusion steps. For the standard DDGM, for simplicity, we use a uniform sampler, while for DAED, we propose a weighted uniform sampler, where the probability of sampling from a given step $t$ is proportional to the given $\beta_t$. This also applies to the Denoising Autoencoder as a part of DAED that is updated accordingly to the new sampler. We update models parameters with AdamW [15] optimizer for a given number of batches as presented in 6. To prevent our model from overfitting, we use dropout [7] with probability $p = 0.3$. Detailed implementation choices, examples of training runs and models can be found in the attached code repository.

Table 6: DDGM and DAED hyperparameters for different datasets

| Dataset | train-steps | depth | channels |
|---|---|---|---|
| FashionMNIST | 100k | 3 | 64, 128, 128 |
| CIFAR10 | 500k | 3 | 128, 256, 256, 256 |
| CelebA | 200k | 4 | 128, 256, 384, 512 |

## A.5 Computational details

Diffusion-based deep generative models are known for being computationally expensive. For our training, we used Nvidia Titan RTX GPUs for complex datasets (CIFAR, CelebA, ImageNet) and Nvidia GeForce 1080Ti for FashionMNIST. Full training of our model on FashionMNIST for 100k steps on a single GPU took approximately 35 hours. For CIFAR and CelebA we used parallel computation based with four GPUs. Full training with this setup took approximately 48 hours. Those estimates are valid for training of both DDGM and DAED.

## A.6 A comparison between DAED and DDGMs with more parameters

The DAED model uses two separate UNet models for the generative and denoising parts. As a result, it has twice as many parameters as a DDGM. In Table 7 we compare DEAD with DDGMs that have a comparable number of parameters. We double the size of the UNet model for vanilla DDGM in two setups. In the first one we increase the number of convolution channels, while in the second one, we double the number of residual blocks.

Table 7: A comparison of DAED with DDGMs of different sizes on the FashionMNIST dataset.

| | Total Params (mln.) | Inference Time (sec. per sample) | FID ↓ | Prec ↑ | Rec ↑ |
|---|---|---|---|---|---|
| DDGM | 8.8 | 0.65 | 7.8 | 72 | 65 |
| DDGM $1.5\times$ channels | 19.8 | 0.84 | 8 | 74 | 65 |
| DDGM $2\times$ blocks | 15.1 | 1.19 | 7.5 | 66 | 66 |
| DAED | 17.6 | 0.66 | 5.7 | 69 | 64 |

The results in Table 7 suggest that the performance of DAED over DDGMs cannot be attributed purely to the larger number of parameters. As we increase the number of layers of the UNet used by the DDGM, we see only a slight improvement of the performance. Furthermore, a larger UNet leads to a significant increase in the inference time compared to the smaller DDGM and DAED.