# OpenReview forum: "On Analyzing Generative and Denoising Capabilities of Diffusion-based Deep Generative Models"
_NeurIPS.cc/2022/Conference — NeurIPS 2022 Accept_

### Official Review · Reviewer_s9uB · 2022-07-07

**Rating:** 3
**Confidence:** 5
**Soundness:** 1 poor
**Presentation:** 1 poor
**Contribution:** 1 poor

**Summary:**

In this work, the authors investigate the generative and denoising capabilities of the Diffusion-based Deep Generative Models (DDGMs), and claimed that a DDGM is a combination of two parts. The first one generates noisy samples from the pure noise by inputting more signal from a learned data distribution, while the second one removes the remaining noise from the signal. Based on the observation, the authors proposed a new approach dubbed DAED that is explicitly built as a combination of a generative component (a DDGM) and a denoising one (a DAE).  Empirical experiments are performed to validate the claim.

**Questions:**

Please explain the relation to the aforementioned reference [1] by Benny et al.  Furthermore, the introduction of the new variational bounds and the denoising autoencoder should be justified as they appear redundnat.

**Ethics Review Area:**

["I don’t know"]

**Limitations:**

Limitations are not stated.

**Strengths And Weaknesses:**

Strength: Various empirical validation of the authors' claim.

Weakness: The approach and observations are not novel. Similar observations are already made in the following work. The experimental verifications appear similar.

[1] Yaniv Benny, Lior Wolf, "Dynamic Dual-Output Diffusion Models", Proceedings of the IEEE/CVF Conference on Computer Vision and Pattern Recognition (CVPR), 2022, pp. 11482-11491

Moreover, the variational lower bound (11) has been already used in the derivation of the DDPM, so it is not clear why the new bound and related discussion are necessary. Furthermore, the way of training score functions is closely related to the denoising author encder. Therefore, the decomposition of the generative parts is denoising parts in DDGMs are trivial.

---

> ### Author Response · Authors · 2022-08-02
> **Response to Reviewer s9uB**
>
> **“Moreover, the variational lower bound (11) has been already used in the derivation of the DDPM, so it is not clear why the new bound and related discussion are necessary. Furthermore, the way of training score functions is closely related to the denoising author encder. Therefore, the decomposition of the generative parts is denoising parts in DDGMs are trivial.”**
>
> In equation (11), we present a calculation of VLB, which is a loss function we use to train our model. We do not claim that this is our contribution, we apply variational inference to turn a DAE into a generative model.
> We disagree that the decomposition of DDGMs into generative and denoising parts is trivial. In our submission, we run a series of novel experiments that were not published in any other work. Agreeing with the view of Prof. Black [[https://perceiving-systems.blog/en/post/novelty-in-science](https://perceiving-systems.blog/en/post/novelty-in-science)], we follow his point of view that a novel paper does not have to be complex, difficult to understand, or surprising to see. The goal of our submission was to shed light on the diffusion process in DDGM through a series of novel experiments and formulation. We are happy to see that other reviewers think that it is a significant observation that will have an impact on the community. We hope that our answer to your review clarifies why.

---

> > ### Comment · Reviewer_s9uB · 2022-08-06
> > **Thanks for the response.**
> >
> > Thanks for the kind reply to my concern. Some of the concerns have been addressed, so I will raise my rating by one point.

---

> > > ### Author Response · Authors · 2022-08-08
> > > **Response to Reviewer s9uB**
> > >
> > > Thank you for your response. We are open an eager to further discussion.

---

> ### Author Response · Authors · 2022-08-02
> **Response to Reviewer s9uB**
>
> **"Please explain the relation to the aforementioned reference [1] by Benny et al."**
> We are thankful for pointing out a very interesting work by Benny and Wolf. However, please take into account that it was presented at the conference 2 months after the deadline for NeurIPS. We gladly acknowledged it as an essential step towards understanding DDGMs, and included it in the related works section of our revised submission. **Moreover, we want to point out that observations by Benny and Wolf are orthogonal to those included in our submission, and the two approaches can in fact benefit from each other.**
>
> In [1] authors focus on the differences between two formulations of DDGMs - one where the model at each stage predicts the original target input image $x_0$ and the opposite one where the model is trained to predict noise $\epsilon$ added to the original input at a given step. Authors observe that at different stages of the diffusion process, it is more beneficial to predict either the first or the second output. On top of this observation, they introduce Dual-Output Diffusion Model that predicts at the same time both targets and interpolation parameter that is used to generate the final prediction as a weighted combination of the two outputs. This allows authors to achieve remarkable results and improvement over the SOTA.
>
> In our approach, we do not explore this aspect of DDGMs, but instead, we focus on the analysis of how the diffusion process (defined to predict noise) changes from the generative part – when new image features appear, to the denoising one – when remaining noise is removed from already formed image. We show experimentally that at the latest ~10% of diffusion steps, the noise removal process is, in fact, agnostic to the original signal from the data, which encourages us to divide it with two separate parametrizations explicitly. For the second model, we show that it can be understood as a single step denoising autoencoder (DAE) that combines the last ~10% of diffusion steps into a single one. However, we do not change the objective of the DAE, and contrary to [1], we always predict the added noise that we remove from the input. The benefits we observe with DAED come solely from the explicit separation of parameters for the generative and denoising models without changing the models’ objectives. Moreover, we believe that thanks to observations from [1], we might be able to further improve our solution by changing the objective of the DDGM part to predict $x_0$ instead of $\epsilon$. At the same time, our research on benefits from separate parametrizations might also be incorporated with [1] and lead to the ultimate solution where both model’s output and parameters change with time.
>
> **“The experimental verifications appear similar.”**
> We disagree with the statement about the significant similarity between experiments run in our submission and work by Benny and Wolf. [1] clearly focuses on experiments explaining how the objective of DDGM affects the diffusion process. In our submission, we evaluate which part of this process leverages training data to generate new image features and which can be described as deterministic noise removal.
>
> The main experiments presented in [1] are:
> - Comparison of how different diffusion losses change over diffusion timesteps
> - Evaluation of differences in the reverse diffusion process for a model trained to predict $\epsilon$ (subtractive) vs. the one predicting $x_0$ (additive)
> - Evaluation of how the interpolation parameter designed to combine outputs from two heads of proposed model changes with time
> - Comparison of the proposed method with different solutions aiming to reduce the number of diffusion steps
> - Experiments on combining the proposed solution with classifier guidance
>
> The main experiments presented in our submission are:
> - Analysis of signal-to-noise ratio and changes of signal-to-noise ratio in the diffusion process
> - Analysis of how reconstruction error to the original sample changes with the number of diffusion steps and how much information from similar data examples is needed for reconstruction
> - Evaluation of how the selection of splitting point between generative DDGM and denoising DAE affect the final performance
> - Comparison of DAED performance with standard DDGM
> - Evaluation of portability of diffusion models and DAED to other datasets

---

### Official Review · Reviewer_4jY9 · 2022-07-09

**Rating:** 7
**Confidence:** 3
**Soundness:** 3 good
**Presentation:** 3 good
**Contribution:** 3 good

**Summary:**

The presented work makes the observation that the reverse diffusion step of DDGMs can be decomposed into two distinct parts: one corresponding to denoising and the other corresponding to generation. Furthermore, the authors extensively study different aspects of such a decomposition and motivated by it, propose a new setup called DAED combining DDGMs with DAE. The proposed DAED seems to have performance on par with DDGMs for most setups while it has a better performance compared to DDGMs regarding generalizabilitry to different data distributions.

**Questions:**

Please refer to the weaknesses described above. It would be nice to hear what the authors think regarding the mentioned weaknesses.

In addition, I would be interested to see some analysis on the diversity of generated samples from the proposed DAED compared to traditional DDGMs.

Finally, I would also be interested to see what impact different noise schedules have on the performance of DAED and how the transition points change with respect to different noising schedules (e.g.- linear, cosine etc.)

**Limitations:**

I enjoyed reading the paper and find it a well written paper with novel observations. Some limitations have been mentioned in the weaknesses section of the review.

**Strengths And Weaknesses:**

***Strengths***
1) The work makes an interesting and novel observation that the generative reverse diffusion step of DDGMs can be decomposed into two separate parts for denoising and generation. This is a significant observation and will help the community better understand DDGMs many aspects of which till date are not fully understood.
2)  The proposal of DAED leverages on the observations made in this paper and is a sensible approach while achieving at par or better performance to traditional DDGMs.

***Weaknesses***
1) The authors show there is a transition point between the denoising and generative capabilities of traditional DDGMs and choosing when to switch from one mode to another affects performance significantly. Can the authors recommend how this transition point can be automatically selected for a given dataset?

2) The proposal of composing a DDGM with DAE is sensible. I am wondering if stacking DAEs (stacked denoising autoencoders) can further help with performance of DAED.

---

> ### Author Response · Authors · 2022-08-02
> **Response to Reviewer 4jY9**
>
> **[1] “The authors show there is a transition point between the denoising and generative capabilities of traditional DDGMs and choosing when to switch from one mode to another affects performance significantly. Can the authors recommend how this transition point can be automatically selected for a given dataset?”**
> We suggest searching for the right step for switching between DDGM and DAE on the basis of the signal-to-noise ratio for a given dataset that can be easily computed before the training. In general, we observe the high performance of the method when DAE is applied on the steps where $log(SNR)>1$. However, as presented in Tab.1-Tab.3. in the appendix, an early switch to DAE affects mostly the diversity (recall) of the generations and not their quality. For specific applications such as real-data simulation, omitting part of the data distribution might not be acceptable. However, for the others, e.g., text-to-speech synthesis, a wide variety of outputs for a given prompt is not needed; hence, an earlier switch between DDGM and DAE might be possible.
>
> **[2] "The proposal of composing a DDGM with DAE is sensible. I am wondering if stacking DAEs (stacked denoising autoencoders) can further help with performance of DAED."**
> Thank you for this comment. In fact, we considered this approach as a method to further improve the performance of the DAED. However, stacking multiple models raises additional research questions: How much noise should we try to remove with each model? What is the suitable models' architecture for such a setup since more models require additional memory? Can we generate multiple DAEs with hypernetwork to limit the memory footprint?
> We will be happy to search for the correct answers to those questions, but for now, they seem to be beyond the scope of this submission. Nevertheless, the proposed idea is an exciting direction for future work.
>
> **[3] “In addition, I would be interested to see some analysis on the diversity of generated samples from the proposed DAED compared to traditional DDGMs.”**
> To better outline the change in samples' diversity generated with DDGM and DAED, we report the distributed recall metric proposed by M. S. Sajjadi Assessing generative models via precision and recall (Neurips 2018). As expected, the more steps we concatenate into a single Denoising pass, the lower the diversity of generated samples, while the precision (quality) remains intact or even slightly rises. Exemplar generations, with all models and for all datasets, that might depict those differences visually are included in the appendix.
>
> **[4] "Finally, I would also be interested to see what impact different noise schedules have on the performance of DAED and how the transition points change with respect to different noising schedules (e.g.- linear, cosine etc.)"**
> Due to the space limitation, we include additional plots of Signal to Noise ratio for different noise schedulers on all datasets in the supplementary material. Interestingly, for the cosine scheduler signal is retained in the input for longer. This means that we can find similar switching points earlier - for approximately 20-30% of the last diffusion steps instead of 10-20% in the linear schedule. We present results of experiments with this setup in Figure 6, where we use pre-trained DDGM from [Nichol 2021] that was trained with a cosine noise schedule.

---

> > ### Comment · Reviewer_4jY9 · 2022-08-08
> > **Thank you for your responses**
> >
> > I thank the authors for their responses to my questions. Given the satisfactory responses, I am happy to maintain my score of 7 and advocate the acceptance of this work.

---

### Official Review · Reviewer_QKFo · 2022-07-10

**Rating:** 6
**Confidence:** 2
**Soundness:** 2 fair
**Presentation:** 3 good
**Contribution:** 2 fair

**Summary:**

The paper investigate the behaviour of diffusion-based deep generative models (DDGM) and argues that DDGM can be separated into a generative and a denoising part. The authors combine DDGMs with denoising auto encoders to introduce DAED and show that this model can improve the overall sample and reconstruction quality.

**Questions:**

- The metrics show precision and recall values for the images. What does this mean?

**Limitations:**

see above.

**Strengths And Weaknesses:**

Pros:
- The paper is well written and easy to follow.
- The interpretation of DDGM as generator and denoiser seems novel and offers interesting insights into the field of diffusion-based models.
- The experimental results seem to indicate improved generative performance.

Cons:
- The authors rely on t/T vs reconstruction to define the denoising and generation stages. However, this is a bit arbitrary and and should be further discussed. Additionally, it would be interesting to show standard deviations for this behaviour to further support this distinguishment.
- The improved results might be due to the increased model capacity instead of an actually better model.

---

> ### Author Response · Authors · 2022-08-02
> **Response to Reviewer QKFo**
>
> **[1]“The metrics show precision and recall values for the images. What does this mean?”**
> Following related works [Nichol 2021, Dharival2021], we evaluate our methods with FID and distributed precision and recall introduced in M. S. Sajjadi Assessing generative models via precision and recall (Neurips 2018). As indicated by the authors, those metrics disentangle the standard FID into two aspects - the quality of generated images (precision) and their diversity (recall). We find it especially useful to outline the strengths and limitations of the DAED formulation proposed in this work. As expected and indicated in Tab.1. of the main article and Tables 1-3 in the appendix, the more steps we concatenate into DAE, the lower the diversity of generated examples (recall), while the quality (precision) improves or remains intact.
>
> **[2]“The improved results might be due to the increased model capacity instead of an actually better model.”**
> To answer this question, we run two additional experiments on the FashionMNIST dataset, doubling the size of the UNet model for vanilla DDGM. In the first setup, we increase the number of convolution channels, while in the second one, we double the number of residual blocks. As presented in Table 7 in the Appendix A.6, (copied below), matching the number of DAED parameters only slightly changes the results.
>
> |                     | Total params (mln) | Inference time (sec. per sample) | FID $\downarrow$ | Prec $\uparrow$ | Rec $\uparrow$ |
> |---------------------|:------------------:|:--------------------------------:|:---:|------|-----|
> |         DDGM        |         8.8        |               0.65               | 7.8 |  72  |  65 |
> | DDGM 1,5 x channels |        19.8        |               0.84               |  8  |  74  |  65 |
> |   DDGM 2 x blocks   |        15.1        |               1,19               | 7.5 |  66  |  66 |
> |        DAED         |        17.6        |               0.66               | 5.7 | 69   | 64  |
>
> Nevertheless, even though the number of total parameters in DAED is doubled, this should not be considered comparable to a basic DDGM with a twice as big model. The most problematic part of DDGM is the time of inference that requires multiple passes through the model. Doubling the number of parameters for DDGM effectively slows down every pass. This is not the case for DAED, where at each step, only a single set of parameters is used with a switch to a DAE at a specific step. Hence, the second - DAE model needs only to be stored in the memory (not even necessary on GPU), which is of much lower cost when compared to doubling the size of the DDGM model. We present inference speed comparison of evaluated methods in the table.
>
> **[3] “The authors rely on t/T vs reconstruction to define the denoising and generation stages. However, this is a bit arbitrary and and should be further discussed. Additionally, it would be interesting to show standard deviations for this behaviour to further support this distinguishment.”**
> We conduct an experiment with t/T vs reconstruction loss to show that a transition point between the generative and denoising parts of the diffusion model exists, while we suggest that the boundary between them is fluid. We further discuss this observation in Sections 6.1 and 6.2. In Tab.1-Tab.3. of supplementary material we provided additional evaluations for different splitting points for DAED. An early switch to DAE affects mostly the diversity (recall) of the generations and not their quality. For specific applications such as real-data simulation, omitting part of the data distribution might not be acceptable. However, for the others, e.g., text-to-speech synthesis, a wide variety of outputs for a given prompt is not needed; hence, an earlier switch between DDGM and DAE might be possible.
> Thank you for pointing out the importance of including the standard deviation for the behaviour we discussed. We include this information in Figure 3 of the updated submission. As we can notice, there is not much divergence and our claims remain valid.

---

> > ### Comment · Reviewer_QKFo · 2022-08-08
> > **Thanks for the response!**
> >
> > Thanks for the response to my questions. I increased my rating by one point.
> >
> > However, one question remains: The additional results indicate that an increase in parameters in the DDGM does not improve the generative performance (only one of the metrics improves, whereas another worsens). Is this related to optimisation issues?

---

> > > ### Author Response · Authors · 2022-08-08
> > > **Response to Reviewer QKFo**
> > >
> > > Thank you for the response. In all of our experiments, we rely on architectural choices from “Improved Denoising Diffusion Probabilistic Models” by Nicholas and Dharival. The UNet model employed in this work is already overparametrized (almost 9 mln parameters) for a simple FashionMNIST dataset with 28x28 images. This is why increasing the model's capacity even further does not impact the model's performance.

---

> > > > ### Comment · Reviewer_QKFo · 2022-08-08
> > > > **Thank you for the clarification**
> > > >
> > > > Thanks a lot for the clarification. I guess I am trying to understand what causes a model to optimise one of Prec/Rec/FID over the others and what that means in practice. Do you have any intuition for this?
> > > >
> > > > Furthermore, did you run any experiments with models that were not as much overparametrised?

---

> > > > > ### Author Response · Authors · 2022-08-09
> > > > > **Response**
> > > > >
> > > > > In our experiments, we did not evaluate how the size of the diffusion model affects the final performance, and unfortunately, we won't be able to provide any meaningful experimental study at the time of this discussion. Nevertheless, we evaluate in detail how combining the different amount of last diffusion steps into a single denoising pass affect the final FID, Precision, and Recall of generations. This might help in gaining intuition on how to understand those metrics.
> > > > > As presented in Tab.1-Tab.3. in the appendix, an early switch to DAE affects mostly the diversity (recall) of the generations and not their quality. This can be observed as increasing smoothness of generated samples, as presented in Figure 5. of the main submission - generated samples are more probable (higher precision) but sometimes lack small details observed in the real data (lower recall). In practice, it might not be acceptable for specific applications such as real-data simulation. However, for the others, e.g., text-to-speech synthesis, a wide variety of outputs for a given prompt is not needed; hence, an earlier switch between DDGM and DAE might be possible.

---

### Official Review · Reviewer_aLmz · 2022-07-12

**Rating:** 7
**Confidence:** 4
**Soundness:** 3 good
**Presentation:** 3 good
**Contribution:** 3 good

**Summary:**

- In this study, the authors observe that score-based denoising diffusion models can actually be decomposed into two phases. They argue that by substituting the first half of the two phases with VAE, higher quality images can be generated. They then propose DAED (Denoising Auto-Encoder + Diffusion), a tandem connection of VAE and Diffusion model.
- The authors examine the fundamentals of the division into these two phases. For example, what level of SNR should be set at the transition point is considered.
- Many other basic issues related to this problem are also carefully considered.


**Questions:**

- It would be nice if the points raised in the Cons section above is clarified.


**Limitations:**

- This study is more interested in understanding the fundamental issues of denoising diffusion models rather than updating SOTA. So the performance is not competitive with the latest SOTA. This is one of the limitations of this study, and is acknowledged by the authors.


**Strengths And Weaknesses:**

### pros

- The authors found that the diffusion model can actually be divided into a first half and a second half, with the first half being used to roughly generate the noisy images, and the second half being used for denoising. The authors also found that solving the first half with VAE is more effective than solving it with diffusion, and they verified this experimentally. The argument is clear and intuitive.
- As can be seen from the loss of the denoising diffusion model, one needs to solve a problem of estimating $\varepsilon'$ from $\varepsilon$ (where $\varepsilon$ and $\varepsilon'$ are very close) when SNR is very low, which is seemingly redundant. It is a natural idea to combine all the first half of the "noise to noise prediction" process into a single module to reduce redundancy, and increase the gain per step. The authors have verified that this is indeed the case.
- The authors' findings also mean that 80-90% of the total diffusion steps can be replaced by a single pass VAE, which would also make the synthesis more efficient.


### cons

- In the usual denoising diffusion models, the parameters are shared by a single model regardless of the log SNR value. However, in the proposed method, separate models are used for the VAE and diffusion. Thus, the number of parameters in the total model would be roughly doubled. This raises the question whether it is fair to compare the proposed tandem method directly with a vanilla denoising diffusion model. The dependence on the total model size should be examined.

---

> ### Author Response · Authors · 2022-08-02
> **Response to Reviewer aLmz**
>
> **“In the usual denoising diffusion models, the parameters are shared by a single model regardless of the log SNR value. However, in the proposed method, separate models are used for the VAE and diffusion. Thus, the number of parameters in the total model would be roughly doubled. This raises the question whether it is fair to compare the proposed tandem method directly with a vanilla denoising diffusion model. The dependence on the total model size should be examined.”**
>
> To answer this question, we run two additional experiments on the FashionMNIST dataset, doubling the size of the UNet model for vanilla DDGM. In the first setup, we increase the number of convolution channels, while in the second one, we double the number of residual blocks. As presented in Table 7 in the Appendix A.6, (copied below), matching the number of DAED parameters only slightly changes the results.
>
> |                     | Total params (mln) | Inference time (sec. per sample) | FID $\downarrow$ | Prec $\uparrow$ | Rec $\uparrow$ |
> |---------------------|:------------------:|:--------------------------------:|:---:|------|-----|
> |         DDGM        |         8.8        |               0.65               | 7.8 |  72  |  65 |
> | DDGM 1,5 x channels |        19.8        |               0.84               |  8  |  74  |  65 |
> |   DDGM 2 x blocks   |        15.1        |               1,19               | 7.5 |  66  |  66 |
> |        DAED         |        17.6        |               0.66               | 5.7 | 69   | 64  |
>
> Nevertheless, even though the number of total parameters in DAED is doubled, this should not be considered comparable to a basic DDGM with a twice as big model. The most problematic part of DDGM is the time of inference that requires multiple passes through the model. Doubling the number of parameters for DDGM effectively slows down every pass. This is not the case for DAED, where at each step, only a single set of parameters is used with a switch to a DAE at a specific step. Hence, the second - DAE model needs only to be stored in the memory (not even necessary on GPU), which is of much lower cost when compared to doubling the size of the DDGM model. We present inference speed comparison of evaluated methods in the table.

---

> > ### Comment · Reviewer_aLmz · 2022-08-08
> > **Thank you for your response**
> >
> > Thank you for answering my concerns.
> >
> > I understood that increasing the capacity of the DDGM would probably only marginally increase the accuracy of score function predictions, and without increasing the number of iterations, the overall quality would not change so much. On the other hand, the proposed method has a qualitative improvement in the framework of the synthesis procedure, and has some advantages over the vanilla iteration. This additional result seems convincing to me.

---

### Author Response · Authors · 2022-08-02
**Comment to all reviewers**

Dear reviewers,

We would like to thank you all for your time and effort in reading and evaluating our paper. We have put a lot of effort into addressing all of your questions and concerns in the comments below. We also uploaded the updated version of the manuscript, where all the changes are marked in orange for your convenience.

---

### Comment · Area_Chair_nzad · 2022-08-07
**Discussion period**

Thank you to all the reviewers for the great effort in reviewing the paper and the authors for the responses.

As the author-reviewer discussion period is almost over, I want to ensure that reviewers have read the authors' responses and engage with the authors if needed.

If you haven't done this, could you please take a moment to read through the authors' responses, update the reviews to indicate that you have read the authors' responses, or communicate with the authors if needed? You can also share in private conversations with the reviewing team.

Please continue to share your thoughts. Thank you!

---

### Meta-Review · Area_Chair_nzad · 2022-08-23

**Recommendation:** Accept
**Confidence:** Less certain

**Metareview:**

The paper analyzes diffusion-based deep generative models (DDGMs). The paper postulates that a DDGMs can be divided into two parts, a denoiser and a generator.

After the rebuttal and discussion period, three out of four reviewers supported acceptance of the paper. The reviewers aLmZ, QKfo, and 4jY9 all find the interpretation of a diffusion-based deep generative model as a decomposition of a generator and denoising part interesting. Those reviewers also note that this interpretation is useful as parts of the diffusion steps can potentially be replaced with a VAE, which would make the synthesis more efficient.

The reviewers aLmZ and QKFo also note that one comparison is perhaps slightly unfair in that the number of parameters of the two models considered was different; the authors cleared this up with new simulations showing that matching the number of parameters (as should be done for a better comparison) does not substantially change the conclusions of the experiment.

Finally, reviewer s9uB finds the approach and observations not to be novel and states that similar observations have been made in a recent CVPR paper by Benny and Wolf. While both the paper by Benny and Wolf and the paper under review study denoising diffusion models, the approaches taken by the two papers are substantially different, and both provide value to the community.

I also find the interpretation of the DDGMs as a denoiser and generator to be interesting and useful and therefore recommend acceptance of the paper.


**Award:**

No

---

### Decision · Program_Chairs · 2022-09-14

Accept